# Polyphenol-Dietary Fiber Conjugates from Fruits and Vegetables: Nature and Biological Fate in a Food and Nutrition Perspective

**DOI:** 10.3390/foods12051052

**Published:** 2023-03-01

**Authors:** Ana Fernandes, Nuno Mateus, Victor de Freitas

**Affiliations:** Laboratório Associado para a Química Verde (LAQV-REQUIMTE), Departamento de Química e Bioquímica, Faculdade de Ciências, Universidade do Porto, Rua do Campo Alegre, s/n, 4169-007 Porto, Portugal

**Keywords:** bioactivity, dietary fibers, extractable polyphenols, functional foods, non-extractable polyphenols, polysaccharides

## Abstract

In the past few years, numerous studies have investigated the correlation between polyphenol intake and the prevention of several chronic diseases. Research regarding the global biological fate and bioactivity has been directed to extractable polyphenols that can be found in aqueous-organic extracts, obtained from plant-derived foods. Nevertheless, significant amounts of non-extractable polyphenols, closely associated with the plant cell wall matrix (namely with dietary fibers), are also delivered during digestion, although they are ignored in biological, nutritional, and epidemiological studies. These conjugates have gained the spotlight because they may exert their bioactivities for much longer than extractable polyphenols. Additionally, from a technological food perspective, polyphenols combined with dietary fibers have become increasingly interesting as they could be useful for the food industry to enhance technological functionalities. Non-extractable polyphenols include low molecular weight compounds such as phenolic acids and high molecular weight polymeric compounds such as proanthocyanidins and hydrolysable tannins. Studies concerning these conjugates are scarce, and usually refer to the compositional analysis of individual components rather than to the whole fraction. In this context, the knowledge and exploitation of non-extractable polyphenol-dietary fiber conjugates will be the focus of this review, aiming to access their potential nutritional and biological effect, together with their functional properties.

## 1. Introduction

In recent years, epidemiological studies and related meta-analysis established an association between the long-term consumption of fruit and vegetable-rich diets and the health benefits towards a vast array of human diseases (e.g., cardiovascular diseases, cancer, chronic inflammation, degenerative diseases, or metabolic disorders such as type II diabetes) [1,2]. The health benefits arise mainly from the non-nutritive bioactives present in fruit and vegetable foodstuffs, commonly named phytochemicals. Among these, polyphenols have stood out as nutraceuticals and functional ingredients [3]. In this sense, the comprehensive knowledge of the total content of polyphenols in foods and diets is a critical step for biological, epidemiological, and clinical studies, addressing their potential health effects [4]. 

Polyphenol absorption on the food matrix has been increasingly understood over the years. The concept of “food matrix” points towards the fact that bioactives are part of a larger complex set of cellular origin (in fruits and vegetables) or even structures produced during food processing, where they may interact at different length scales [5,6]. Due to these interactions, an appreciable amount of non-extractable polyphenols can still remain in the solid residues, absorbed to dietary fibers [7]. Although non-extractable polyphenols are an important fraction in plant-based foods, they are usually neglected in bioavailability and metabolism studies, as well as in clinical trials and nutritional studies. Additionally, the therapeutic use of non-extractable polyphenol-dietary fiber conjugates through diet is missing, particularly related to the topics of bioaccessibility, pharmacokinetics, and bioavailability of polyphenols, which could influence the real use of these bioactives by the human organism. While extractable polyphenols are dissolved in the stomach and small intestine where they can be partially absorbed, non-extractable polyphenols cannot be absorbed at the small intestine [8]. They reach the lower gastrointestinal tract almost intact in association with the vegetable cell wall. There, they exert systematic bioactive effects before or after being catabolized by colonic microbiota, yielding different metabolites that may counteract the effects of dietary pro-oxidants, and promoting colonic homeostasis [9,10,11]. These compounds can also modulate the gut microbiota, resulting in a healthier profile that could be a potential tool to counteract several chronic diseases related to intestinal dysbiosis such as diabetes, obesity, and inflammatory bowel diseases [10,12].

In this context, research priority should be directed towards the non-extractable fraction in plant-derived foods as well as in agri-food wastes and by-products, promoting their integral valorization and reincorporation to the food supply chain, as an innovative material that combines the properties of both polyphenols and dietary fibers.

This review summarizes the relevance of polyphenol-dietary fiber conjugates in a food technology and health perspective, as many of the pathways sustaining the potential of these conjugates remain largely unexplored, including, for instance, their bioavailability, metabolism in the human digestive track, or microbiota modulatory potential. This review will lay the foundations for future studies aiming to unravel the applicability of non-extractable polyphenols in functional foods development and in nutraceutical fields as dietary supplements.

## 2. Polyphenols 

Polyphenols are secondary plant metabolites with fundamental roles in plant physiology and morphology [13]. They can be found in fruits, vegetables, and cereals, thus representing an essential part of the human daily diet. In plant-based foodstuff, polyphenols can be associated with major sensorial properties, such as color, flavor, and taste (e.g., sweet and bitter taste), and to astringency perception [14,15,16]. More notably, polyphenols have been in the spotlight of recent research exploiting their potential as functional ingredients in diverse food systems due to their therapeutic and health promoting properties.

The general structure of polyphenols includes two aromatic nuclei with one or more hydroxyl substituent. They are commonly divided into two main classes, the flavonoids, characterized by a C_6_-C_3_-C_6_ flavanic core, and the nonflavonoids with a C_1_-C_6_, C_3_-C_6_ or a C_6_-C_2_-C_6_ core. Flavonoids account for nearly two-thirds of dietary polyphenols and, depending on the type of heterocycle present and its substitution pattern, they may be subdivided into anthocyanins, flavonols, flavan-3-ols, flavones, flavanones, isoflavones, and chalcones [3]. Phenolic acids (hydroxycinnamic and hydroxybenzoic acids) and stilbenes are amongst the most common dietary non-flavonoids [17] (Figure 1). Moreover, polyphenols are also present in the polymeric form in plants. Lignins are polymers of monolignols, such as *p*-coumaric and sinapic acid, while tannins may exist as hydrolysable and condensed tannins (or proanthocyanidins). Hydrolysable and condensed tannins are made of saccharide esters derived from gallic/ellagic acids or flavan-3-ols units, respectively (Figure 2), and they are particularly important as they can attain high molecular weights and complex polymeric structures [18].

Although polyphenols can be found practically in all plant-derived foodstuffs, their distribution and quantities are extremely variable [19], ranging from more than 15 g per 100 g in cloves to 7.8 mg per 100 mL in rosé wine [20]. Other external factors can also regulate the polyphenolic content in plants and derived foodstuff, such as genetic, environmental (e.g., type of soil, sun exposure, stage of ripeness), and technological factors (e.g., industrial processing, storage, or culinary preparation) [21,22]. Furthermore, their total content in fruits and vegetables are, most of the time, underestimated as they only include extractable polyphenols that can be easily recovered with aqueous-organic solvents. Compounds entrapped within the plant cell walls (non-extractable polyphenols or macromolecular antioxidants) are usually overlooked and not even considered in nutritional, clinical, and epidemiological studies, resulting in great dissimilarities between the estimated and ingested content [4,23]. 

## 3. Extractable and Non-Extractable Polyphenols

Polyphenols can be found either in extractable (soluble-free) and non-extractable form (insoluble or bound), distributed in various tissues/organs of the plant body [24]. The concept of extractable vs. non-extractable polyphenols is related to their extractability in aqueous and organic solvents, although non-extractable polyphenols can also be released and extracted after proper chemical or enzymatic hydrolysis [25]. Most of the extractable polyphenols are located in the vacuoles of plant cells and do not interact physically or chemically with other plant macromolecules, being easily extracted after plant tissue disruption with polar aqueous/organic solvents [14,26]. Flavonoids, phenolic acids, stilbenes, and lignans are some of the phenolics that can be found in extractable form.

Non-extractable polyphenols, including high-molecular weight proanthocyanins, hydrolysable tannins, flavonoids, and low-molecular weight phenolics, can be virtually found in all types of plant-derived foods [27]. Non-extractable polyphenols can be found cross-linked to cell wall structural components such as cellulose, pectin, hemicellulose (e.g., arabinoxylans), lignin, and structural proteins through covalent bonds (ester bonds with hydroxyl groups of cell wall substances via carboxylic groups, ether linkages with the aromatic hydroxyl groups, or by C-C bonds) [24,28]. These phytochemicals play a major role in the connection of cell wall substances, enhancing cell wall mechanical strength and structural rigidity, being also involved in the protection against UV radiation and harmful organisms (pathogens, insects, and herbivores) [29,30], with antibacterial, antifungal, and antioxidant functions.

Phenolic acids are the most common bound phenolic compounds in natural sources. Hydroxycinnamic acids (e.g., ferulic acid) are covalently linked to arabinogalactans in sugar beet [31] or spinach [32], as well as to arabinoxylans in wheat [33], bamboo [34], and maize [35]. Additionally, polysaccharides can form covalent bonds with polyphenols during food processing [36]. For instance, heat treatment and acid conditions may cause polyphenol depolymerization and the formation of carbocations, being able to randomly react with cell wall nucleophilic compounds through covalent bonds [37,38,39]. In addition, oxidation of polyphenols in damaged tissues may occur [39].

However, it should be emphasized that the concept of non-extractable polyphenols is a broader concept than the term “bound phenolics”, which is usually associated with phenolic acids covalently linked to cell walls [7]. The type of chemical interactions between non-extractable polyphenols and dietary fibers also includes the formation of ordered junctions stabilized by arrays of non-covalent interactions (e.g., hydrogen bonding, ionic bond, electrostatic interaction, and hydrophobic effect), during or after physical damage and the senescence of plants [40]. Furthermore, polyphenols may be physically entrapped in various cellular structures or entrapped by a surface adsorption phenomenon or by encapsulation in hydrophobic pockets within the biopolymer network and porous structures (particularly for dietary fibers) embedded into intact cellular structures [14,24]. This mechanism is particularly notorious with pectic polysaccharides due to their higher structural flexibility [41,42]. As these bonds are individually weak, these interactions are stable only above a minimum critical length, and their formation and disruption often occur as sharp and cooperative processes [43]. 

### 3.1. Types and Content of Non-Extractable Polyphenols in Foods 

Non-extractable polyphenols include several classes of polyphenols, either polymeric or single, linked to plant macromolecules, and can be found in fruits, vegetables, legumes, cereal grains, and seeds. In fruits and vegetables, non-extractable polyphenols encompass an average of 24% of the total amount [4,44,45]. For instance, banana (*Musa acuminate*), orange (*citrus sinensis*), and apple (*Malus domestica*) have nearly 33, 24, and 7% of non-extractable polyphenols, respectively [44]. Carrots (*Daucus carota*) and onions (*Allium cepa*) have about 38 and 10% of non-extractable polyphenols, respectively [46,47]. On the other hand, cereals, such as brown rice, present 80–90% of total polyphenols in the bound form [48]. Plant-based food by-products, such as peals, pomace, and seeds, also possess high amounts of non-extractable polyphenols. Cranberry pomace (*Vaccinium macrocarpon*) has approximately 76% of non-extractable polyphenols [49]. 

Phenolic acids (mostly ferulic acid derivatives) are the most abundant non-extractable polyphenols, being covalently bound (by ester, ether, or C-C bonds) to the cell wall matrix [50]. Other phenolic acids may be found to be associated with cell wall polysaccharides such as *p*-hydroxyphenyl, syringyl, or *p*-coumaroyl moieties. They are mostly found in cereals but are also present in other plant-based foodstuffs such as spinach and sugar beet (ferulic and *p*-coumaric acids), orange flavedo (ferulic, sinapic, and *p*-coumaric acids), and lentils (*p*-coumaric acid) [51].

Proanthocyanidins with a higher degree of polymerization can be found in the non-extractable fractions, including both proanthocyanidins bound to cell wall polysaccharides and proteins. Interactions between proanthocyanidins and cell walls occur mostly through non-covalent binding, including hydrogen bonds, hydrophobic effect, or van der Waals forces. It is becoming clear that the adsorption mechanism between proanthocyanidins and cell wall constituents is affected by a multitude of factors, including environmental parameters (e.g., pH, temperature, ionic strength) and the physicochemical properties of the interacting partners: morphology (surface area, porosity, pore shape), chemical composition (sugar ratio, solubility, branching complexity), molecular weight, and polyphenol molecular architecture (polymerization degree, contributing to the increase of hydroxyl and aryl groups available to form hydrogen bonds and to establish hydrophobic interactions, respectively, degree of methylation and acetylation, degree of galloylation/hydroxylation, conformation) [40]. Furthermore, higher proportions of pectic polysaccharides impart higher flexibility to the structure, thus allowing for a higher contact surface area, while higher proportions of lignin and cellulose, with higher structural rigidity, allow for fewer adsorption interactions [39,52].

Most hydrolysable tannins have been detected in water-organic extracts, although significant amounts can also be found in the resulting solid residues [27]. The way by which hydrolysable tannins are associated with the food matrix has not yet been established. In addition, no information about individual tannins was obtained, as the current method for their determination causes tannin depolymerization [53,54,55].

Flavonols, including rutin, isoquercitrin, quercitrin, and quercetin were also found in the non-extractable fraction of tropical or subtropical fruits or leaves, tomato peel (also flavanones in this source), wine, beer, *Hibiscus sabdariffa* flowers, and tropical and subtropical fruits or leaves [56,57,58,59,60]. However, the exact nature of the interactions between flavonoids and the solid cell wall matrix has not been properly addressed. Regarding anthocyanins, studies are somehow controversial as the data reported for the non-extractable anthocyanins determined after acid hydrolysis may correspond to the hydrolysis of proanthocyanidins [61,62]. Thus, the presence of a fraction of non-extractable anthocyanidins remains to be properly established. Additionally, there are several limitations in the methodology for the extraction of these pigments from the food matrix. For instance, the use of glycosidases or acid hydrolysis to release anthocyanins may cause the formation of less stable aglycones [56]. In a previous work, an anthocyanidin molecule (malvidin aglycone) associated with macromolecular material has been reported [63], suggesting, as reported for melanoidins, the covalent linkage of anthocyanins to the polymeric material [64], specially to pectic polysaccharides. 

In this sense, non-extractable polyphenols represent a very important fraction to be considered in a nutritional and health perspective. They may be slowly and continuously released in the human gastrointestinal tract and during colonic fermentation, which can improve bioaccessibility and potential bioavailability and exert high bioactivity on tissues and cells for a longer time. 

### 3.2. Release of Non-Extractable Polyphenols from Dietary Fibers

Although non-extractable polyphenols can be regarded as an important fraction of phenolic compounds in plant-based foods [7], the challenge is that their chemical composition analysis requires several isolation steps, as they cannot be detected with the usual analytical procedures for either fibers or extractable polyphenols (e.g., HPLC analysis). However, the extraction methods are often not systematically optimized and greatly dependent on the chemical and physical nature of the food plants [65], giving rise to different recovery yields. Furthermore, the accuracy of the results may be questionable, as some of the methodologies employed to release and assess non-extractable polyphenols are often destructive and inefficient. Depending on the chosen methodology, on the extraction methodology order applied, or on the plant matrix, degradation or incomplete release may occur, resulting in underestimated values [25].

Extraction methods to recover non-extractable polyphenols from the cell matrix for further analyzes in the corresponding hydrolysates include chemical (acid or alkali) [28,66], physical (e.g., microwave-, ultrasound-assisted hydrolysis, pressurized solvent/liquid extractions, far-infrared (FIR) radiation-assisted, or pulsed electric field-assisted) [67,68,69,70,71,72,73], or enzymatic hydrolysis, with carbohydrate-hydrolyzing enzymes such as cellulases, hemicellulases, pectinases, proteases, glucanases, or bacterial enzymes [74,75]. While alkaline hydrolysis is more effective in promoting the disruption of both ester and ether bonds, linking polyphenols to cell wall constituents [66], acid hydrolysis mainly promotes the disruption of glycosidic bonds, generally leaving ester bonds intact. Compared to acid or alkaline hydrolysis, enzymatic hydrolysis allows a lower loss of phenolic compounds as a moderate pH is used during the extraction procedure. Additionally, carbohydrate-hydrolyzing enzymes may also induce the production of aglycone moieties due to the presence of β-glucosidase, β-galactosidase, or α-L-arabinoside activities, which can be detrimental to the stability of several flavonoids, such as anthocyanins [76]. In addition to the carbohydrases, esterases have also been shown to be efficient in the release of non-extractable polyphenols [77].

Given the specificity of each hydrolytic system, a single methodology is probably inadequate to perform the full assessment of non-extractable polyphenols. Thus, a combination of different hydrolytic systems may provide more complete information regarding the non-extractable fraction.

Food processes such as fermentation an germination, as well as thermomechanical processes such as extrusion, have also been shown to be effective non-thermal food processing methods for the release of non-extractable phenolics from the cell wall matrix [78,79]. Abdel-Aty, 2019 [80] reported the improved phenolic content, antioxidant and antimicrobial activities of garden cress seeds, using solid-state fermentation. However, other works have also reported the decrease in the levels of phenolic compounds, which can be attributed to the degradation and hydrolysis of phenolic compounds [81]. Several works have shown the increased content of non-extractable phenolics due to germination [82,83]. This bioprocess induces the activation of cell metabolism, which results in the release of hydrolytic enzymes, thus affecting phenolic compound content [84]. 

Other food processes such as roasting, extrusion, or boiling have shown potential to release phenolic compounds associated with cell walls [85,86]. These thermal processes involve the use of high temperatures and/or pressure, leading to the disruption of the cell wall matrix of foods through the depolymerization of pectins and hemicelluloses, thus enhancing the release of bound phenolics. On the other hand, the high energy extrusion conditions degrade bioactive phenolic compounds, with a consequent reduction of the antioxidant capacity [87,88,89,90]. Hydrothermal treatments such as boiling have also been shown to cause chemical reactions between phenolic compounds and other compounds, such as proteins, forming irreversible covalent bonds that are not hydrolyzed during the extraction process [91,92].

## 4. Biological Activity of Non-Extractable Polyphenol-Dietary Fibers

Over the past few years, several biological activities have been attributed to non-extractable polyphenols (either isolated or rich matrices), combining in vitro models, cell cultures, animal models, or human trials [7]. However, it is important to note that non-extractable polyphenols have been previously subjected to a hydrolytic process to release polyphenolic compounds from the carbohydrate or protein moiety, following extraction and analysis of specific phenolic compounds in the corresponding hydrolysates and the assessment of their bioactivities similarly to extractable polyphenols [25]. For instance, antioxidant capacity, antiproliferative and apoptotic effect on cancer cells, inhibition of carbohydrate and lipid metabolism enzymes, or anti-inflammatory activity are some of the bioactivities that have been described [93,94,95,96]. Additionally, the released single phenolic acids and flavonoids have been extensively examined for their biological activities against diabetes, cardiovascular and neurodegenerative disease, and cancers in cell lines and in vivo models [97,98,99]. 

After ingestion, a significant part of the non-extractable polyphenols reaches the colon, playing a major role in the reduction of local oxidative stress and boosting anti-inflammatory mechanisms and immunity through a chemical action in the gut environment [100]. On the other hand, besides acting as carriers for polyphenols, thus affecting their bioaccessibility and potential bioavailability [41], cell wall polysaccharides can also influence host wellbeing and health through a variety of mechanisms, depending on their dietary source, physicochemical structure, fermentability, and physiological properties in the gut [101], modulating intestinal microflora, improving gastrointestinal health, and regulating brain signals, affecting the gut–brain axis. There is also evidence suggesting that polyphenol-dietary fiber interaction can modulate the fermentation of polyphenols in the gut [102,103,104]. Dietary fibers are also associated with favorable body weight and overall metabolic health, being also associated with a reduced risk for the development of cardiovascular disease [105], some forms of cancer [106], and depression [107,108]. 

However, the specific biological activity of non-extractable polyphenol-dietary fiber conjugates is not well established due to the insoluble nature of these compounds. For instance, the measurement of the antioxidant activity has been limited most of the time to soluble materials, with the extraction procedure being considered a critical step [8]. However, many of the insoluble components cannot be solubilized without altering their molecular nature by chemical or enzymatic treatment. Thus, direct methodologies to assess the antioxidant activity of insoluble material have been developed [109,110]. The QUENCHER methodology has been shown to accurately measure the antioxidant capacity of antioxidants (e.g., ABTS, DPPH, FRAP, ORAC, CUPRAC) bound to insoluble matrices [111,112,113,114], without extraction and hydrolysis processes, by a surface reaction phenomenon. The QUENCHER concept was also adapted to the methodologies that evaluate the scavenging capacity of superoxide, hydroxyl, and lipid peroxyl radicals [115]. On the other hand, all antioxidant compounds are present in a dynamic system where radicals and antioxidant compounds react continuously with each other, with the coexistence of multiple antioxidants possibly resulting in additive, synergistic, or antagonistic interactions [116]. Previous studies have shown that the antioxidant activity of non-extractable antioxidants bound to dietary fibers can be regenerated by extractable antioxidants in the liquid phase, as the extractable ones can provide electrons or hydrogen atoms to the formers, regenerating them [117,118]. Consequently, non-extractable ingredients with an increased antioxidant capacity can be obtained, as during the digestion process, they can react with the free radicals and at the same time being regenerated by extractable compounds [111], a promising framework in the development of functional foods.

Furthermore, non-extractable polyphenols and dietary fibers have shown to positively affect SCFA production, with rats fed with dietary fibers enriched with polyphenols extracts showing a higher SCFA production compared with only fibers [119]. Mice supplemented with apple pomace flour showed a lower body weight gain and improved glucose tolerance [120]. The daily intake of grape pomace consumed in bread, biscuits, or directly mixed with water improved blood pressure, glycaemia, postprandial insulin, and antioxidant defense [121]. Finally, its consumption in burgers supplemented with wine pomace improved fasting glucose, insulin resistance, plasma antioxidant levels, and oxidative damage markers [122]. The same pomace contributed to a decrease on inflammation markers [123]. 

## 5. Non-Extractable Polyphenol-Dietary Fiber Fate through the Human Gut

Bioavailability of a compound can be defined as the fraction that can reach systematic circulation after administration, so that they can exert bioactivity [124]. Thus, bioavailability represents one of the most relevant aspects for the potential therapeutic effects of the ingested compounds. With respect to in vivo potential, non-extractable polyphenols are essentially considered non-bioavailable as they can reach the colon still covalently linked or chemically adsorbed or entangled within the fiber matrix, with their absorption mechanisms in the upper part of the gastrointestinal tract greatly depending on their release from the food matrix. From a nutritional point of view, non-extractable polyphenols are a relevant group of compounds for two main reasons: (a) most of the non-extractable polyphenols are not affected by the acidic conditions at the gastric phase or by the enzymes of small intestine [111]. Only a small percentage (about 5–10%), can be partially released from the food matrix, namely the water soluble non-extractable polyphenols, due to their poor structure. In this case, polyphenols may be absorbed through the small intestine mucosa by direct solubilization in the intestinal fluids in physiological conditions (37 °C, pH 1–7, mobility, transit time) and after ester bond cleavage, by the action of mucosa cell esterases [28]. Glucuronidation, sulfation, or methylation may also occur at the level of intestinal mucosa [125,126]; (b) the remaining non-absorbed polyphenols, connected to cell wall macromolecules, can reach the lower gut intact, where they play a major role in the reduction of local oxidative stress and in modifying the microbiota composition, thus improving gut permeability and boosting the anti-inflammatory mechanisms and immunity [100]. There, they undergo fermentation by the colonic microbiota, releasing absorbable metabolites or by the action of some intestinal enzymes able to break covalent bonds, such as esterases [127,128]. Colonic microorganisms, including, for instance, *Bifidobacterium* spp., *Clostridium* spp., and *Lactobacillus* spp., secrete a variety of extracellular enzymes such as carbohydrolases, protease, and other types of enzymes, leading to the disruption of the cell wall matrix and hydrolysis of covalent bonds or release of entangled polyphenols, and producing metabolites. Additionally, colonic microbiota cleaves the glycosidic linkage and breaks down the flavonoid structure of the unabsorbed phenolics, releasing aglycones as well as converting them into small molecules [129]. 

On the other hand, the action of the microbiome on macromolecules, mainly carbohydrate and protein, produces mainly short-chain fatty acids (SCFA) (acetic, propionic, butyric) and gases from carbohydrates, along with nitrogen compounds from proteins and low molecular weight phenolic compounds and some phenolic metabolites (hydroxyphenylacetic, hydroxyphenylvaleric, and hydroxyphenylpropionic acids or urolithin) from non-extractable and also from extractable phenolic compounds that were not absorbed in the small intestine and which have been implicated in the suppression of intestinal inflammation and improvement of the intestinal microenvironment [130]. 

The released polyphenols (aglycones and small molecules) in the gut lumen render a myriad of health benefits that influence the fermentation environment of the colon by decreasing pH, enhancing the intestinal antioxidant status, which may protect against dietary prooxidants and free radical and prevent the growth of cancer-inducing microorganisms [28,131,132]. Once free in the gut lumen, polyphenols and their metabolites can pass the colon mucosa and be absorbed into the bloodstream, through passive diffusion and/or active transport by transporters, such as glucose transporters, ATP-binding cassette transporters, and monocarboxylic acid transporters in the intestinal epithelium [133]. Through the portal vein, polyphenols arrive in the liver where they are mainly metabolized as glucuronides metabolites and, to a minor extent, as sulfated and methylated compounds. These metabolites may return to the digestive tube through the bile, or pass into the bloodstream, reaching tissues and organs, allowing protection against chronic diseases [12] (Figure 3). Colonic phenolic acids may also influence the regulation of the immune system response at the epithelial level or modulate colonic microbiome, through the activation of SCFA excretion, promoting gut health, thus presenting a prebiotic effect [134]. On the other hand, SCFA have also been shown to affect and to enhance the uptake of phenolic metabolites [135].

The metabolites produced by colonic fermentation from the free and non-extractable polyphenols during digestion were shown to be similar. However, the specific characteristics of non-extractable polyphenols, such as high molecular weight and/or association with other macromolecules present in the food matrix, cause two specific features in the colonic transformation. From the delay of polyphenol levels observed in the plasma after intake, comparing non-extractable-rich matrices and free phenolics-rich matrices, it could be concluded that the metabolites derived from non-extractable polyphenols circulate for longer periods in the human body than those produced from free phenolics, exerting their effects much longer than extractable polyphenols in a living organism [130,136]. In fact, non-extractable polyphenols and other phenolic metabolites may persist in the human plasma for up to 3–4 days after ingestion, playing key bioactive roles, such as modulators of low-grade inflammation and cell-signaling pathway mediators [8,137]. According to Vitaglione et al., 2008, the slow and continuous release of non-extractable polyphenols may favorably act in vivo, quenching the soluble radicals that are continuously formed in the intestinal tract, as opposed to the extractable polyphenols that are immediately absorbed and metabolized in the gastrointestinal tract [8], thus allowing a continuous protection. In this sense, dietary fibers may act as natural delivery carriers, improving not only polyphenol chemical stability and solubility but also modulating its bioavailability throughout the gastrointestinal tract [138]. In sum, the presence of macromolecular components in non-extractable polyphenols might affect polyphenol bioaccessibility in the small intestine, by decreasing it. At the same time, they may potentially increase the polyphenol amount that reaches the lower parts of the digestive tract [139], being released and bio-transformed into catabolites. 

## 6. Non-Extractable Polyphenol-Dietary Fibers: A Potential Functional Ingredient

Single polyphenols and dietary fibers have been widely used as functional ingredients in foodstuffs due to their well-recognized physiological roles. For instance, dietary fibers have been associated with an increase of the volume of fecal bulk, decrease of the intestinal transit time, reduction of cholesterol and postprandial blood sugar levels, stimulation of intestinal microflora proliferation, such as *Bifidobacterium* (bacteria associated with colon, stomach, breast, and prostate cancer prevention [140]), and promote the formation of SCFA, reducing luminal pH, which helps to prevent colonization and infections from pathogenic bacteria [141]. Moreover, some bioactive polysaccharides possess other health-promoting properties (e.g., antimicrobial, antitumoral, and immunostimulating effects) [142]. These bioactivities are mainly associated with polysaccharide physicochemical and conformational properties [143].

On the other hand, polyphenols are attracting more attention due to their antioxidant capacity and antimicrobial, anti-carcinogenic, and anti-inflammatory activity in the human body [144]. They also exhibit an inhibition effect on digestive enzymes, such as lipase, amylase, and glucosidase [63,145]. Polyphenols can also act as prebiotics due to their inhibitory effect against pathogens and stimulation effects on beneficial bacteria [134,146]. 

From a technological perspective, dietary fibers may affect the technological properties of food related to texture, due to water and fat-holding ability, gelling and swelling capacity, emulsion stability, increased viscosity, and foam capacity and stability, as well as solvent retention capacities (e.g., lactic acid, sodium carbonate, sucrose) [147,148]. The viscosity of dietary fibers is due to physical interaction between fiber particles, strongly associated with fiber microstructure [149]. Additionally, they may be used as non-caloric bulking agents for partial replacement of flour, fat, or sugar. Polyphenols, on the other hand, have been used as natural dyes, preservatives in food formulations (antimicrobial, antioxidants, and antibacterial activity), favoring, thickening agents, and prebiotic ingredients [66,150]. In this sense, non-extractable polyphenols associated with dietary fibers have assumed an increasingly significance, as they hold the promise to provide physiological and technological properties (e.g., texture, color, health benefits, or stability during shelf-life) of both substances in a single material [144,151]. 

Hence, the search for innovative non-extractable polyphenol-dietary fiber materials to the food industry has been emerging, also representing an opportunity to apply sustainable circular economy models. However, it should also be taken into consideration that the supramolecular structure and physical properties of dietary fibers (e.g., rheology, encapsulation efficiency, water solubility, and structural stability) can be altered through polyphenol interaction [152,153]. For instance, previous studies have shown that polyphenols may induce a viscosity decrease and the pseudoplastic (Newtonian) behavior of polysaccharide solutions through the formation of polysaccharide–polyphenol aggregates (e.g., β-glucan, galactomannan, guar, and xanthan gum) [154,155]. Another study showed that phenolic acids and flavonoids can alter the gelatinization, retrogradation, and digestibility of starch. Procyanidins, on the other hand, can increase the crystallinity, bonding temperature, and maximum viscosity of starch [156]. 

In a food concept, non-extractable polyphenol-dietary fibers can be defined as a dietary fiber concentrate with significant amounts of associated natural polyphenols [157]. Furthermore, there are certain requirements that must be observed so that the material can be considered as a potential food ingredient, namely the dietary fiber content should be higher than 50% in a dry weight basis and the antioxidant capacity must be an intrinsic property of the material, deriving from the natural constituents and should not be related to added antioxidants or any other constituents released through chemical or enzymatic treatment of the original components. Regarding the antioxidant capacity, 1000 mg of antioxidant dietary fibers should inhibit lipid oxidation equivalent to at least 200 mg of vitamin E and present a free radical scavenging capacity equivalent to at least 50 mg of vitamin E [144,158]. 

The fruit and vegetable industry generates high amounts of wastes and by-products that can be reused in different applications. Multifunctional polyphenol-dietary fiber ingredients, with higher health-promoting effects (source of bioactive compounds, such as vitamins, minerals, and polyphenolic compounds) and technological functionalities represents a suitable alternative to valorize these residues. For instance, olive pomace has been reported as an antioxidant, antimicrobial, and prebiotic ingredient, with potential beneficial effects on the gut microbiota and with functional properties (solubility and water and oil-holding capacity) [159,160]. The most commonly prepared fruit and vegetable dietary fibers with associated polyphenols that could be employed as sources to develop functional ingredients are obtained from guava, açai and mango fruits, chokeberries, apples, cranberries, carrot peels, pineapple shells, cocoa, and cabbage leaves, or they are by-products of *Vitis vinifera* L. grapes (pomaces) or apple pomace [161,162,163,164,165] (Table 1). Cereals, particularly bran and husk, are also a rich source, with a high proportion of non-extractable polyphenols associated with dietary fibers [166].

Polyphenol-dietary fiber materials are available in various forms, such as granules, seeds, hulls, shells, or grains, as well as in the form of powdered ingredient [183]. Typically, these preparations are subjected to a washing and stabilization step (normally by heat treatment) to inactivate microorganisms and enzymes [184]. Decontamination with sodium hypochlorite solution, followed by rinsing with water, can be applied to guarantee food safety [185]. Depending on the vegetable source, separation of liquid and solid fractions may be required, which allows a reduction of the drying time and the caramelization extent of the free sugars during drying [186]. Additionally, this process allows one to obtain a solid residue with lower water content and to collect a liquid part rich in water soluble compounds, allowing a wider valorization of the raw material. Typically, these preparations have been prepared with low water activity, obtained after drying and grinding. These two parameters can affect the techno-functional properties of the powdered ingredient. While drying temperature can be managed to increase phenolic compounds and consequent health benefits of the by-products, grinding and the particle size affect the technological properties of the flours, particularly the hydration properties. At lower particle size, dietary fiber water holding capacity, water retention capacity, and swelling capacity increase. However, further decreases lead to a decrease on the hydration properties, possibly due to soluble and insoluble fiber content [187]. 

In recent years, the presence of a specific type of naturally occurring water-soluble polyphenol-polysaccharide conjugates has been described in the flowering parts and leaves of many traditional medicinal plants [188] and some food crops (e.g., grape pomace, lingonberry, or rice), with these conjugates being the focus of several studies. Despite their relatively lower purity and higher heterogeneity, some common and characteristic structural features (consisting of two parts, a polyphenolic and a polysaccharide moiety) and biological activities have been identified. Purification methods are similar to those applied for polysaccharides due to their macromolecular nature, including anion exchange chromatography and gel filtration chromatography. In general, polyphenol glycoconjugates exhibit several biological activities, including anticoagulant, antioxidant, radioprotective, anti-platelet, antitussive, and bronchodilatory activity [189,190,191,192,193,194]. More notably, naturally occurring polyphenol-polysaccharide conjugates were also shown to have a bright prospect in the field of food and nutraceuticals, presenting an antiglycation and antihiperglycemic effect, digestive enzymes inhibitory effect, anti-obesity, and regulation of gut microbiota [63,195,196,197]. Glucose transport in Caco-2 cells was also found to be significantly inhibited by water soluble polyphenol-polysaccharide treatments [63]. Recovery of these conjugates through conventional solvent extraction techniques (hot water or alkaline extraction) [189] require long extraction time, high energy consumption, and organic solvents, which may cause negative effects on human health upon ingestion due to their residue in the final product. Among modern extraction methods, subcritical water extraction (SWE), pressurized liquid extraction (PLE), ultrasound-assisted extraction (UAE), and microwave-assisted extraction (MAE) have shown the potential to replace conventional extraction techniques [198,199].

Besides naturally occurring polyphenol-dietary fibers, synthesized conjugates may also play a fundamental role in the development of new functional ingredients, with improved physicochemical and bioactive properties, compared to the unmodified polyphenols and polysaccharides. Phenolic acids, including caffeic, gallic, ferulic, vanillic, and coumaric acids, have been covalently grafted onto chitosan [200], starch [201], curdlan [202], pectin [203], and gum arabic [204]. Other flavonoids, such as catechin, rutin, quercetin, hesperidin, curcumin, naringenin, and proanthocyanindins, have also been successfully grafted onto polysaccharides [205,206,207,208,209]. The grafting reactions can be achieved by carbodiimide-mediated coupling reaction, free-radical induced reaction, or enzyme-mediated polyphenols-polysaccharide conjugation reactions (e.g., laccase, tyrosinase, horseradish peroxidase, and choroperoxidase) [210]. Wheat bran, a source of dietary fibers, was shown to reacted with polyphenols from different sources (green and black tea and white and red wine) under alkaline conditions with free amino groups available on the surface of wheat bran, followed by polymerization of the soluble antioxidants [211]. Synthetic conjugates have also shown several bioactive properties, such as antioxidant, anticarcinogenic, anti-inflammatory, and antidiabetic activity, which could allow their application in the food industry, such as in food preservation and packaging or in active nanocarriers for delivery of active ingredients [211]. Thus, understanding the synthetic strategy, purification methodology, and the knowledge of the functional and physiological properties of polyphenol-polysaccharide conjugates is fundamental for the development of innovative food ingredients.

## 7. Non-Extractable Polyphenol-Dietary Fibers: An Approach in Food Industry and Innovation Potential

Non-extractable polyphenol-dietary fibers are promising ingredients to bring innovation and functionality to the food industry, affecting polyphenol bioavailability and promoting high bioactivity on tissues and cells, while potentiating the technological functionalities of both dietary fibers and polyphenols. In fact, non-extractable-dietary fiber conjugates can be seen as innovative delivery systems, acting as polyphenol carriers, and at the same time modulating gut microbiota [7]. On the other hand, using food by-products and wastes to obtain new ingredients that can be re-introduced into the industrial chains can be considered as a solution to mitigate the economic, environmental, and social impact caused by food wastes, bringing innovation to the food sector [65]. 

After the validation of the potential of non-extractable polyphenol-dietary fiber conjugates as functional ingredients, there are some aspects that must be globally considered in the next steps of the development of food products, particularly production processes (as it may impact the texturizing properties), raw-material availability, stability, and storage conditions, including shelf-life, microbiological safety regarding the presence of pesticides and other contaminants from agriculture, and physical hazards. Finally, it should be taken into consideration that the valorization of these bioactive compounds should occur at competitive prices, so that they can be used [212].

Non-extractable polyphenol-dietary fibers from fruits and vegetables have been incorporated into food products, preferably into cereals, as this material presents a better nutritional profile due to the significant content of associated bioactive compounds, lower phytic acid content, and caloric value. Additionally, this material presents a balanced dietary fiber composition (higher fiber content, IDF:SDF ratio) and technological functionalities, namely as thickeners, gelling agents, fillers, and as water and oil holding retainers, as well as in the production of edible films [213,214]. 

Literature has shown the application of this ingredient in different food products, enhancing the bioactive compound level (antioxidants and dietary fiber) in finished food products, and increasing their technological behavior [215]. In bakery products, dried fruit and vegetable pomaces can be added to replace flour, sugar, or fat, thus reducing energy load, while enhancing fiber and antioxidant contents. On the other hand, techno-functional interactions may occur, affecting the physicochemical and sensory properties of the finished product [216]. For instance, pasta fortified with mango peel powder showed a dietary fiber and polyphenol increase of about two and four times, respectively, resulting in an improved storage stability without changing the textural, cooking, and sensory attributes [163]. In addition, the replacement of 20% of wheat flour with mango peel in soft dough biscuits led to a phenolic content increase of up to eight times, compared to the control [183]. Cakes prepared with watermelon and melon peel powders, as a dietary fiber rich in phenolic compounds, showed improved nutritional and antioxidant properties [217], while muffins have been successfully fortified with dried apple peel [218]. Probiotic yogurt fortified with 1% fiber-rich pineapple peel powder was produced, remarkably reducing the fermentation time of milk. However, a loss in firmness and storage modulus could be observed [219]. Other examples of non-extractable polyphenol-dietary fiber application can be found in a large variety of food products, including meat products, bread, cereals, pasta, dressings, cookies, biscuits, and extruded or dairy products [183,219,220,221]. 

## 8. Conclusions and Future Perspectives

The increasing awareness of lifestyle in human well-being has resulted in a great interest in the health-promoting aspects of food. In this sense, the food industry needs to translate nutritional information into consumer demand through the development of natural food products that provide not only appealing sensorial features and gastronomic innovation but also nutritional and health benefits. 

In recent years, the attention on polyphenols research regarding human health benefits has expanded from extractable compounds to non-extractable ones. Non-extractable polyphenol-dietary fiber conjugates are a promising natural bio-ingredient, which have been increasingly recognized as important players in food development, nutrition, and human health. There is a new trend and opportunity to investigate more deeply how non-extractable polyphenols bound to dietary fibers are implicated in the prevention or evolution of different chronic diseases and how they can be used in the development of innovative functional foods. Their application as a global entity, having dual properties, shows a great potential for how functional food development can be used to produce innovative food products with improved techno-functional properties. Moreover, these ingredients can be used to improve the nutritional content of food products or be targeted to consumers with a high risk of human health problems (e.g., type II diabetes, obesity, or cardiovascular disease) or to enable personalized nutrition. 

Although non-extractable polyphenol-dietary fibers present high potential applications for the development of innovative functional ingredients and foods, contributing at the same time to the valorization and exploitation of agri-food by-products and wastes for non-extractable polyphenol-dietary fibers recovery, they have not been implemented on a big scale. Future research should be directed towards the global exploitation of plant-based by-products, which could lead to the generation of new business, revenues streams, and jobs. Thus, the focus should be addressed towards updating the methodological approach for the obtainment of non-extractable polyphenol-dietary fibers in a multidisciplinary and innovative design. An improved understanding regarding this fraction may allow the establishment of a new bridge between their chemical composition, bioavailability, and their ultimate biological function, contributing for the design of sensorial and texture appealing, healthier, and sustainable foods, with this opening up new possibilities for the food industry to address consumer demands.

## Figures and Tables

**Figure 1 foods-12-01052-f001:**
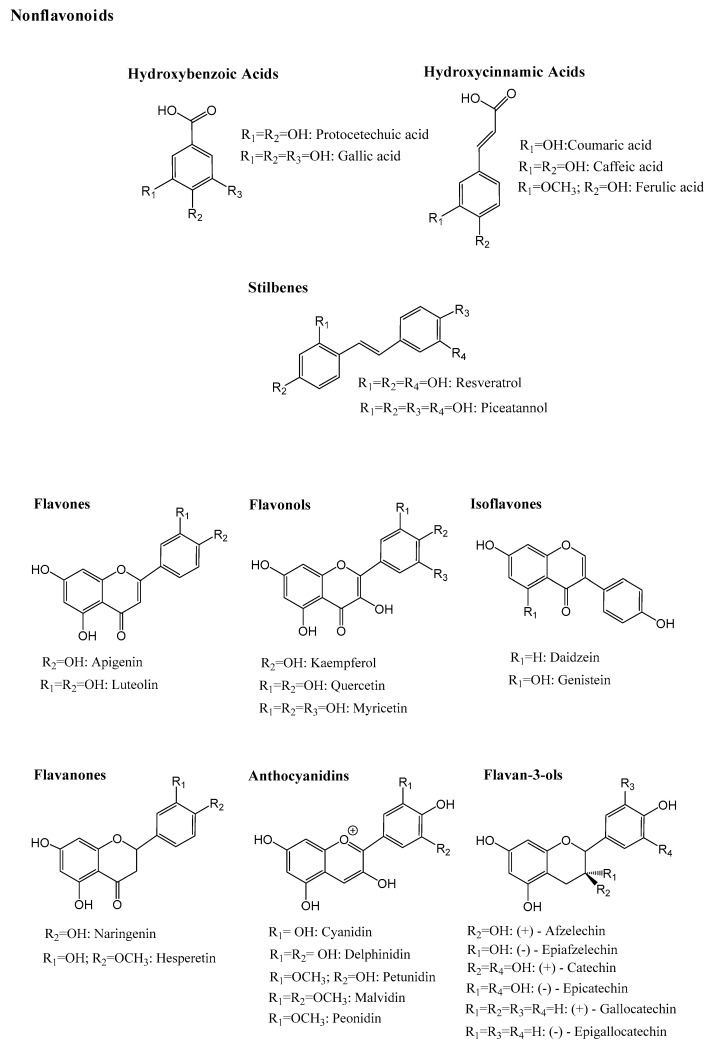
Chemical structure of the major nonflavonoids and flavonoids subclasses (aglycones).

**Figure 2 foods-12-01052-f002:**
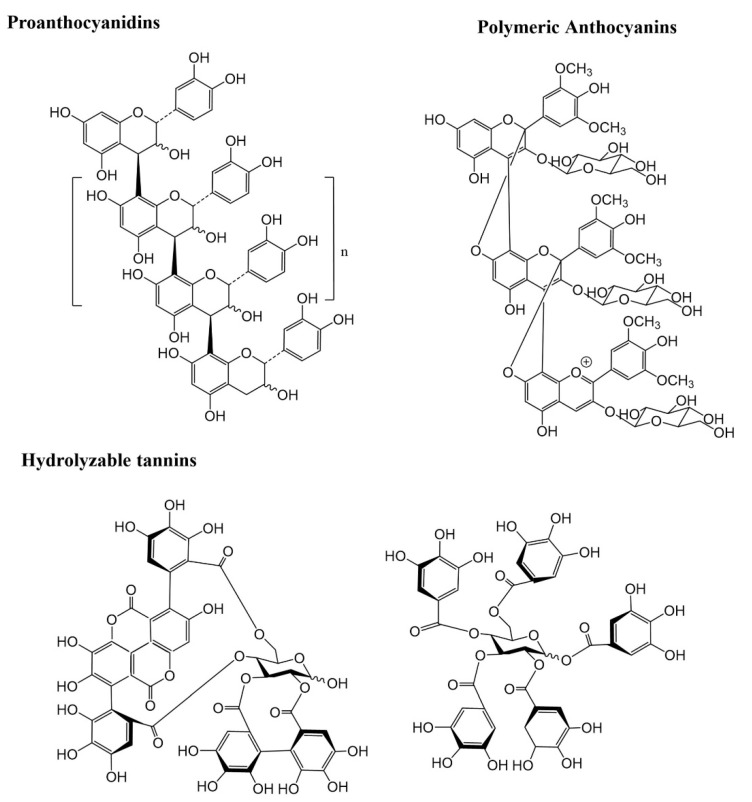
Chemical structure of some representative polymeric phenolic compounds.

**Figure 3 foods-12-01052-f003:**
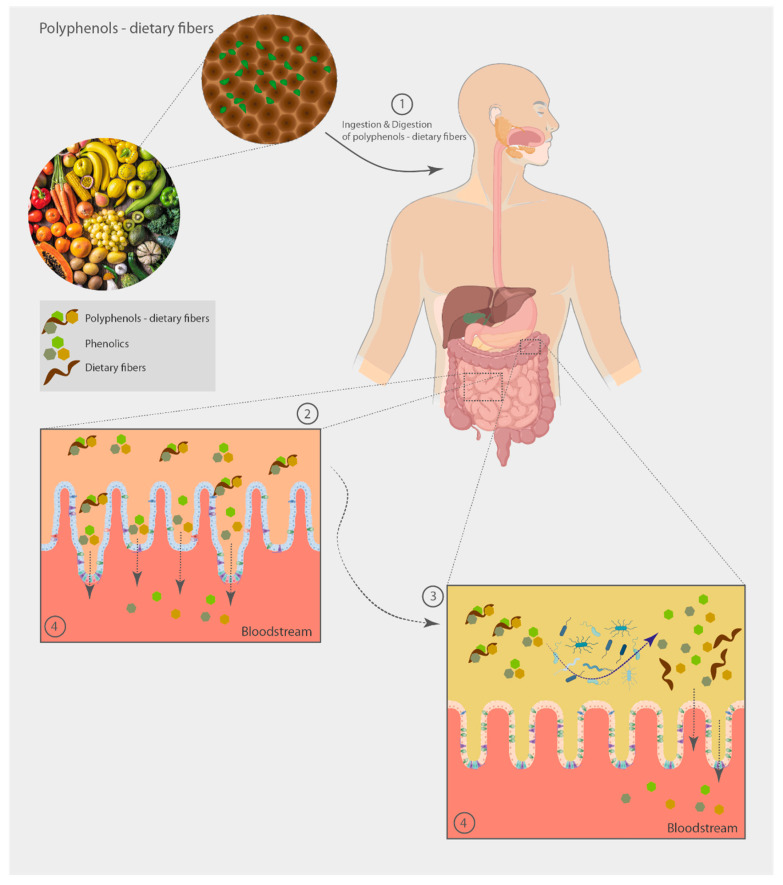
Overview of the metabolic fate of polyphenol-dietary fiber conjugates through the gut. 1—polyphenols can be ingested both in free form and conjugated to the dietary fibers components of plant cell walls; 2—only a small percentage of polyphenol-dietary fiber conjugates can be absorbed through the small intestine mucosa. Others reach the lower gut intact, reducing the local oxidative stress and modifying the microbiota composition; 3—non-absorbed polyphenol-dietary fiber conjugates reach the lower gut where they are fermented by the colonic microbiota, releasing absorbable polyphenols and producing absorbable metabolites. The action of the microbiome on dietary fibers produces mainly SCFA and gases; 4—polyphenols and their metabolites can be absorbed into the bloodstream, through passive diffusion and/or active transport, reaching tissues and organs and exerting biological activities.

**Table 1 foods-12-01052-t001:** Examples of polyphenol-dietary fibers from fruit and vegetable by-products. Adapted from [158].

By-Product	Total Dietary Fiber	Total Phenolic Content	Reference
	(%)	(mg GAE/100 g)	
Mango (*Mangifera indica* L.)	54.2	2170 (TPC)	[167]
peel flour			
Cabbage powder	36.7	322 (TAE)	[168]
Guava (*Psidium guajava*) peel	48.55	5871 (TEP)	[162]
Guava (*Psidium acutangulum*) peel	51.5	5870 (TEP)	[162]
Açaí (*Euterpe oleraceae*) pulp	71.2	1500 (TEP)	[169]
Red grape pomace.	74.0	5.63 (TPC)	[170]
Apple pomace	51.1	1016 (TPC)	[171]
Blueberry pomace powder	26.2	28.514 (TPC)	[172]
Carrot (*Daucus carota*) peels	45.4	1371 (TPC)	[164]
Banana (peels)	41.6	7168.5 (TPC)	[173]
Coffee (pulp, husk, silver, skin,	28.0–80.0	1020–1480 (TPC)	
and spent coffee)			
Grape (*Vitis vinifera* L.) pomace	74.5	2630 (TEP)	[174]
Orange (*Citrus aurantium*) peel	33.1–36.5	0.51 (TPC)	[175]
Orange by-product	71.6	40.7 *	[176]
(albedo,flavedo, and pulp)			
Plantain peel flour	37.6	771 (TEP)	[177]
*Viburnum opulus* (fruits)	38.4	3730 (TPC)	[178]
Cacao pod husk products	59.0	6893 (SD)	[161]
Mexican Blackberry	44.3	4016 (TPC)	[179]
(Rubus fruticosus) residues			
Passion fruit seeds (DCF)	81.5	41.2 *	[180]
Pineapple (DFC)	75.8	129 *	[181]
Papaya pulp (DFC)	59.8	0.47 *	[182]
Tomato peel	86.1	158.1 (TPC)	[57]

GAE, gallic acid equivalent; TPC, total phenolic compounds; TAE, tannic acid equivalents; TEP, total extractable phenolics; SD, soluble phenolics; DFC, dietary fibers concentrate; * mg GAE/g.

## Data Availability

Not applicable.

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
