# Peer review of "Polyphenol-Dietary Fiber Conjugates from Fruits and Vegetables: Nature and Biological Fate in a Food and Nutrition Perspective"

_foods, 2023, doi:10.3390/foods12051052_

Round 1

Reviewer 1 Report

Polyphenols have gained the spotlight because they may exert their bioactivities much longer than extractable polyphenols. Additionally, from a technological food perspective, polyphenols combined with dietary fibers have become increasingly interesting as they could be useful for the food industry to enhance technological functionalities. Non-extractable polyphenols include low molecular weight compounds such as phenolic acids and high molecular weight polymeric compounds like proanthocyanidins and hydrolysable tannins, and they can be found in large amounts in agri-food by-products and wastes. Studies concerning the non-extractable fraction are scarce, and usually refer to the compositional analysis of individual components, rather than to the whole fraction. In this context, the knowledge and exploitation of non-extractable polyphenols-dietary fibers conjugates will be the focus of this review, aiming to access their potential nutritional and biological effect, together with their functional properties.

Overall, the manuscript is interesting and the topic about polyphenols-dietary fibers conjugates is meaningful. However, there still have some issues need to revise.

1.      It is better to limited to plant- original food for the title. “Polyphenols-dietary fibers conjugates from agri-food wastes and by-products: nature and biological fate in a food and nutrition perspective”.

2.      Line 40-42 “The concept of “food matrix” points towards the fact that bioactives are part of a larger complex set of cellular origin (in fruits and vegetables) or even structures produced during food processing, where they may interact at different length scales.”In fact the wholegrain benefit refers to that concept, which benefit the health(Whole grain benefit: oat β-glucan and phenolic compounds synergistically regulates hyperlipidemia via gut microbiota in high-fat-diet mice. Food & Function, 2022, 13(24), 12686-12696. Doi: 10.1039/d2fo01746f.). The phenolic acid in oat play an important role in regulate metabolic syndrome (Oat phenolic compounds regulate metabolic syndrome in high fat diet-fed mice via gut microbiota. Food Bioscience. 50(2022)101946. Doi: 10.1016/j.fbio.2022.101946). However, the

3.      Line50-55. “While extractable polyphenols are dissolved in the stomach and small intestine where they can be partially absorbed, matrix-bound or non-extractable polyphenols cannot be absorbed at the small intestine [6]. They reach the lower gastrointestinal tract almost intact in association with vegetable cell-wall. There, they exert systematic bioactive effects before or after being catabolized by colonic microbiota, yielding different metabolites that may counteract the effects of dietary pro-oxidants, and promoting colonic homeostasis [7,8]” Polyphenols can be divided into free phenolic acids and combined phenolic acids, among them combined phenolic acids play an important role in antioxidative and prebiotic effect(The positive correlation of antioxidant activity and prebiotic effect about oat phenolic compounds. Food Chemistry, 402(2023): 134231.).

4.      “3. Extractable and non-extractable polyphenols”. The Citrus fruits are rich in pectin, phenolics, essential oils, and other bioactive. Pectin, as the main substance in the by-product (e.g., peel and pomace), has potential anti-inflammatory and immunomodulatory effects (Citrus pectin research advances: Derived as a biomaterial in the construction and applications of micro/nano-delivery systems. Food Hydrocolloids,133(2022): 107910.).

5.      “4. Biological activity of non-extractable polyphenols-dietary fibers”. It’s better to modify and refer this article (Recent advances of stimuli-responsive polysaccharide hydrogels in delivery systems: A review. Journal of Agricultural and Food Chemistry. 70(21):6300-6316.).

6.      “7. Non-extractable polyphenols-dietary fibers: an approach in food industry and innovation potential”. The biggest potential utilization is to improve the bioavailability and regulate the gut microbiota.

7.      The reference should be updated.

8.      The grammar issues should be checked.

Author Response

Reviewer #1:

Polyphenols have gained the spotlight because they may exert their bioactivities much longer than extractable polyphenols. Additionally, from a technological food perspective, polyphenols combined with dietary fibers have become increasingly interesting as they could be useful for the food industry to enhance technological functionalities. Non-extractable polyphenols include low molecular weight compounds such as phenolic acids and high molecular weight polymeric compounds like proanthocyanidins and hydrolysable tannins, and they can be found in large amounts in agri-food by-products and wastes. Studies concerning the non-extractable fraction are scarce, and usually refer to the compositional analysis of individual components, rather than to the whole fraction. In this context, the knowledge and exploitation of non-extractable polyphenols-dietary fibers conjugates will be the focus of this review, aiming to access their potential nutritional and biological effect, together with their functional properties.

Overall, the manuscript is interesting and the topic about polyphenols-dietary fibers conjugates is meaningful. However, there still have some issues need to revise.

AUTHORS: Thank you very much for your kind comment.

  1. It is better to limited to plant- original food for the title. “Polyphenols-dietary fibers conjugates from agri-food wastes and by-products: nature and biological fate in a food and nutrition perspective”.

AUTHORS: The original title was modified.

  1. Line 40-42 “The concept of “food matrix” points towards the fact that bioactives are part of a larger complex set of cellular origin (in fruits and vegetables) or even structures produced during food processing, where they may interact at different length scales.”In fact the wholegrain benefit refers to that concept, which benefit the health(Whole grain benefit: oat β-glucan and phenolic compounds synergistically regulates hyperlipidemia via gut microbiota in high-fat-diet mice. Food & Function, 2022, 13(24), 12686-12696. Doi: 10.1039/d2fo01746f.). The phenolic acid in oat play an important role in regulate metabolic syndrome (Oat phenolic compounds regulate metabolic syndrome in high fat diet-fed mice via gut microbiota. Food Bioscience. 50(2022)101946. Doi: 10.1016/j.fbio.2022.101946). However, the

AUTHORS: We apologize, but we couldn´t understand reviewers’ intention. However, we added the references suggested by reviewer, as those references are related to the discussed topic.

  1. Line50-55. “While extractable polyphenols are dissolved in the stomach and small intestine where they can be partially absorbed, matrix-bound or non-extractable polyphenols cannot be absorbed at the small intestine [6]. They reach the lower gastrointestinal tract almost intact in association with vegetable cell-wall. There, they exert systematic bioactive effects before or after being catabolized by colonic microbiota, yielding different metabolites that may counteract the effects of dietary pro-oxidants, and promoting colonic homeostasis [7,8]” Polyphenols can be divided into free phenolic acids and combined phenolic acids, among them combined phenolic acids play an important role in antioxidative and prebiotic effect(The positive correlation of antioxidant activity and prebiotic effect about oat phenolic compounds. Food Chemistry, 402(2023):).

AUTHORS: Suggested reference was added.

  1. “3. Extractable and non-extractable polyphenols”. The Citrus fruits are rich in pectin, phenolics, essential oils, and other bioactive. Pectin, as the main substance in the by-product (e.g., peel and pomace), has potential anti-inflammatory and immunomodulatory effects (Citrus pectin research advances: Derived as a biomaterial in the construction and applications of micro/nano-delivery systems.Food Hydrocolloids,133(2022): ).

AUTHORS: We appreciate suggestion that was made by the reviewer. However, in authors opinion, the previous added references are meaningful.

  1. “4. Biological activity of non-extractable polyphenols-dietary fibers”. It’s better to modify and refer this article (Recent advances of stimuli-responsive polysaccharide hydrogels in delivery systems: A review. Journal of Agricultural and Food Chemistry. 70(21):6300-6316.).

AUTHORS: We appreciate suggestion that was made by the reviewer. However, in authors opinion, the previous added references are meaningful.

  1. “7. Non-extractable polyphenols-dietary fibers: an approach in food industry and innovation potential”. The biggest potential utilization is to improve the bioavailability and regulate the gut microbiota.

AUTHORS: We appreciate reviewers’ suggestion. The text was revised, considering this suggestion (line # 614-620).

  1. The reference should be updated.

  AUTHORS: References were updated.

  1. The grammar issues should be checked.

         AUTHORS: Grammar issues were corrected.

Reviewer 2 Report

This review focuses on polyphenol-dietary fiber conjugates, which comprehensively lie in food and food products, especially in by-products, and obtain great attention in recent years for their health benefits. The topic is an interesting and hot. However, the whole frame should be reorganized, and revisions are needed, questions and suggestions are as following.

1.  Sections 1-3 are the introductions about non-extractable polyphenols, which is generally conjugated with dietary fiber. This part is too expatiatory, and most of this information had been summarized in other reviews about bound phenolics. Sections 4-7 are the major content and attractive to readers, could be re arranged and supplement more important information, for example, the isolation, preparations and structures of polyphenol-dietary fiber conjugates from food or their by-products.

2. Some references should consider the newly published papers, for example, references 49-50, because this topic is very hot. Some newly released papers are suggested, DOI: 10.1016/j.lwt.2021.112611. In addition, some new references maybe are missing, for example, line 76-78. One latest related paper could be referred to, DOI: 10.1016/j.foodchem.2020.127879.

3. Line 84, “et al” should be added after “isoflavones”, flavonoids include more 6 classes mentioned in section.

4. Line 524-535 and table 1 are about polyphenols-dietary fibers from  fruit and vegetable, but we know cereals and pulses, especially their bran and husk, generally contain more bound polyphenols and correspondingly higher polyphenols-dietary fibers than fruits. Therefore, similar table about polyphenols-dietary fibers from cereals and pulses by-products is suggested to be provided.

5. Section 7, “the application of this ingredient in different food products”, this section are mainly about applications of foods or by-products,     which could contain polyphenols-dietary fibers.  Are there any applictions about the isolated polyphenols-dietary fibers?

Author Response

Reviewer #2:

This review focuses on polyphenol-dietary fiber conjugates, which comprehensively lie in food and food products, especially in by-products, and obtain great attention in recent years for their health benefits. The topic is an interesting and hot. However, the whole frame should be reorganized, and revisions are needed, questions and suggestions are as following.

AUTHORS: Thank you very much for your kind comments.

  1. Sections 1-3 are the introductions about non-extractable polyphenols, which is generally conjugated with dietary fiber. This part is too expatiatory, and most of this information had been summarized in other reviews about bound phenolics. Sections 4-7 are the major content and attractive to readers, could be re arranged and supplement more important information, for example, the isolation, preparations and structures of polyphenol-dietary fiber conjugates from food or their by-products.

AUTHORS: Sections 1-3 were thoroughly revised, and some information was removed from the manuscript. Isolation, preparation and structural characterization of polyphenols-dietary fibers conjugates was not the main focus of this review, and for this reason only a brief section was made. However, taking in consideration reviewer comment, we added more detailed information, regarding this topic.

  1. Some references should consider the newly published papers, for example, references 49-50, because this topic is very hot. Some newly released papers are suggested, DOI: 10.1016/j.lwt.2021.112611. In addition, some new references maybe are missing, for example, line 76-78. One latest related paper could be referred to, DOI: 10.1016/j.foodchem.2020.127879.

 AUTHORS: reference considering newly published papers were added.

  1. Line 84, “et al” should be added after “isoflavones”, flavonoids include more 6 classes mentioned in section.

 AUTHORS: correction was made.

  1. Line 524-535 and table 1 are about polyphenols-dietary fibers from  fruit and vegetable, but we know cereals and pulses, especially their bran and husk, generally contain more bound polyphenols and correspondingly higher polyphenols-dietary fibers than fruits. Therefore, similar table about polyphenols-dietary fibers from cereals and pulses by-products is suggested to be provided.

AUTHORS: We appreciate the suggestion. However, fruits and vegetables were the focus of this review and for this reason, Table 1 contained only information regarding these sources. Information related to polyphenols-dietary fibers from cereal and pulses was added (line #558-559).

  1. Section 7, “the application of this ingredient in different food products”, this section are mainly about applications of foods or by-products, which could contain polyphenols-dietary fibers.  Are there any applictions about the isolated polyphenols-dietary fibers?

AUTHORS: To our knowledge, there are no application of naturally occurring isolated polyphenols-dietary fibers in food production. There are several manuscripts related to their bioactive characterization, instead. Produced synthetic conjugates have been applied in the development of active packaging or food preservation.

Reviewer 3 Report

The work presents an interesting topic such as polyphenols that are associated with various food matrices. The paper is well organized and clear in its approach and development of the manuscript.

It does not present important weaknesses, here are some opportunities for improvement:

- The authors of the paper should include at least one e-mail contact.

- L44 Review if the term "strongly absorbed" is the most adequate to describe the relationship between the substance and the matrix that hosts it. 

- The document would improve considerably if resources such as more tables, diagrams, and graphs were added to consolidate the information, as it is written it is a bit heavy to read.

Author Response

Reviewer #3:

The work presents an interesting topic such as polyphenols that are associated with various food matrices. The paper is well organized and clear in its approach and development of the manuscript.

It does not present important weaknesses, here are some opportunities for improvement:

- The authors of the paper should include at least one e-mail contact.

- L44 Review if the term "strongly absorbed" is the most adequate to describe the relationship between the substance and the matrix that hosts it. 

- The document would improve considerably if resources such as more tables, diagrams, and graphs were added to consolidate the information, as it is written it is a bit heavy to read.

AUTHORS: Thank you very much for your kind comments. E-mail of contact was added for all authors. The authors removed the term “strongly absorbed”.

We appreciate reviewer suggestion. The manuscript was thoroughly revised aiming to improve manuscript readability.

Round 2

Reviewer 1 Report

The author has responded the reviewer's comment very well.

The topic is interesting. However, considering about the polyphenols digestibility and metabolic rate, it is better to discuss the polyphenols-dietary fibers conjugate combine with the delivery system, such as stimuli-responsive polysaccharide hydrogels in delivery systems (doi: 10.1021/acs.jafc.2c01080).

Author Response

Dear Jessie Xu,

Thank you very much for the feedback regarding the manuscript Foods-2225536.

The authors have read the comment made by Reviewer #1 and the section was revised. Track-changes tool was used in the manuscript.

Best regards,

Ana Fernandes

(corresponding author)
